# The History and Development of the Percutaneous Pedicle Screw (PPS) System

**DOI:** 10.3390/medicina58081064

**Published:** 2022-08-07

**Authors:** Ken Ishii, Haruki Funao, Norihiro Isogai, Takanori Saito, Takeshi Arizono, Masahiro Hoshino, Koji Sato

**Affiliations:** 1Department of Orthopaedic Surgery, School of Medicine, International University of Health and Welfare (IUHW), Chiba 286-8686, Japan; 2Spine and Spinal Cord Center and Department of Orthopaedic Surgery, International University of Health and Welfare (IUHW) Mita Hospital, Tokyo 108-8329, Japan; 3Department of Orthopaedic Surgery, International University of Health and Welfare (IUHW) Narita Hospital, Chiba 286-8520, Japan; 4Department of Orthopaedic Surgery, Kansai Medical University, Osaka 573-1191, Japan; 5Department of Orthopaedic Surgery, Kyushu Central Hospital, Fukuoka 815-8588, Japan; 6Department of Orthopaedic Surgery, Sonoda Medical Institute Tokyo Spine Center, Tokyo 121-0807, Japan; 7Department of Orthopaedic Surgery, Japanese Red Cross Aichi Medical Center Nagoya Daini Hospital, Nagoya 466-8650, Japan

**Keywords:** minimally invasive spinal treatment (mist), minimally invasive spine surgery (miss), minimally invasive spinal stabilization (mist), percutaneous pedicle screws (pps), spinal instrumentation

## Abstract

Minimally invasive transforaminal lumbar interbody fusion (MIS-TLIF) using the SEXTANT^®^ system (Medtronic) featured the first generation of commercially available percutaneous pedicle screw (PPS) system in 2001. The innovative system has since become standard instrumentation used worldwide, and PPS is now used for long-segment minimally invasive surgery (MIS) spinal fixation from the thoracic spine to the pelvis for pathological conditions. PPS systems have been developed for approximately 20 years for the purpose of improving minimally invasive techniques, safety of instrumentation, and ease of use. The third-generation PPS systems established the insertion technique, and the development of the fourth-generation PPS systems have made great strides in minimizing the number of steps in the operative procedure. In the future, PPS systems are expected to continue making use of the latest technological advancements and to develop further with the aim of ensuring greater safety, reducing operator stress, and preventing complications such as insertion errors and infection. In this review article, we describe the historical evolution from the first-generation PPS system to the current PPS systems used today.

## 1. Introduction

In 1982, Magerl [1] reported the first percutaneous pedicle screw insertion procedure for external fixation. Although Mathews and Long [2] reported a fully percutaneous pedicle screw (PPS) system in 1995, this PPS system was not popularized due to its rod connection that remained subcutaneous and numerous cases of nonunion that resulted from its weak fixation. Then, Foley et al. [3] reported a minimally invasive transforaminal lumbar interbody fusion (MIS-TLIF) using the SEXTANT^®^ system (Medtronic, Memphis, TN, USA) (Figure 1a) in 2001, which was the first generation of commercially available PPS systems. The innovative system has since become standard instrumentation used worldwide and has subsequently provided a foundation for current PPS systems and the minimally invasive spine stabilization (MISt) procedure. The original PPS system was mainly used for 1–2 intervertebral fusion in the simple cases with lumbar degenerative diseases [4]; however, with the increase in screw variations and further development of instrumentation, PPS is now used for long-segment minimally invasive surgery (MIS) spinal fixation from the thoracic spine to the pelvis for pathological conditions such as spinal fractures [5], scoliosis [6], metastasis [7,8], osteoporotic vertebral fracture [9], and discitis/pyogenic spondylosis [10].

Approximately 20 years have passed since the PPS system was reported in 2001, and many improvements and developments have since been achieved to introduce various PPS systems into clinical practice. The PPS system was initially made available with the first-generation SEXTANT^®^ system and was succeeded by the second-generation PathFinder^®^ (Abbott Spine, Austin, TX, USA) (Figure 1b), VIPER^®^ (DePuy Synthes Spine, Raynham, MA, USA) (Figure 1C), MANTIS^®^ (Stryker, Kalamazoo, MI, USA) (Figure 1e) systems, etc. The instrumentation later evolved into a third-generation design, which integrated an extended tab that is in wide use today. In recent years, fourth-generation PPS systems have also been introduced that allow safe placement without the use of conventional guide wires. The PPS insertion technique has become one of the standard procedures for spinal instrumentation. According to the North American Spine Society, the percentage of PPS out of all pedicle screws in the United States has steadily increased from 13% in 2006, 23% in 2009, 36% in 2012, 44% in 2014, and finally reached more than half at 52% in 2016. In Japan, where PPS was introduced in 2005, the PPS system has become widespread due to patient needs and referrals of elderly patients (≥90 years). The percentage of PPS used for spinal fixation in Japan was 42.5 % in 2017, 43.4% in 2018, 45.1% in 2019, and reached approximately half at 44.2% in 2020 (data from Yano Research Institute Ltd., Tokyo, Japan). The market is expected to continue expanding in the future.

## 2. The History and Development of the PPS System

The introduction of the PPS system in clinical practice dates back to 2005 with the SEXTANT^®^ system, which corresponds to the first-generation PPS system. This was succeeded by the following second-generation PPS systems: PathFinder^®^ in 2007; VIPER^®^, MANTIS^®^, SpiRIT^®^ (Synthes, Solothum, Switzerland), and Ballista^®^ (Biomet Spine, Broomfield, CO, USA) in 2009; and ILLICO SE^®^ (Alphatec Spine, Carlsbad, CA, USA) in 2011 (Figure 1d) [11]. These systems were subsequently followed by third-generation PPS systems with an extended tab to easily facilitate slippage correction, which account for most of the currently available PPS systems on the market. Typical third-generation systems include ES2^®^ (Stryker) (Figure 1f), VIPER2 X-tab^®^ (DePuy Synthes Spine) (Figure 2a–d), Voyager^®^ (Medtronic) (Figure 2e,f), PRECEPT^®^ (NuVasive, San Diego, CA, USA) (Figure 3a,b), RELINE^®^ (NuVasive) (Figure 3c,d), Associa Harp^®^/Associa Zique^®^ (KYOCERA, Kyoto, Japan) (Figure 3e,f), CREO^®^ (Globus Medical, Audubon, PA, USA) (Figure 4a,b), Saccura^®^ (Teijin Nakashima Medical, Tokyo, Japan) (Figure 4c), and IBIS^®^/Pisces^®^ (Japan Medical Dynamic Marketing, Tokyo, Japan) (Figure 4d). Furthermore, fourth-generation PPS systems that do not require the use of conventional guidewires were introduced from 2019 onwards, including VIPER PRIME^®^ (DePuy Synthes Spine) (Figure 5a,b) and Voyager ATMAS^®^ (Medtronic) (Figure 5c,d). To date, approximately 30 types of systems have been made available since 2022 when including product revisions (Table 1). The common features of PPS systems introduced to date are described below.

The shape and characteristics of each PPS system are different, but the concept of PPS insertion is common to all systems: PPS is inserted percutaneously under fluoroscopy or computer-assisted navigation in order to minimize damage to the surrounding soft tissues [11]. Early PPS systems presented various problems, including their limited ability to correct spinal slippage and their heavy, easily detachable, and complicated extender assemblies that were attached to the screw head. In addition, early PPS systems were also limited to one-segment MIS-TLIF. The PathFinder^®^ system (Figure 1b) later introduced a simplified system over the SEXTANT^®^, which resolved the problems caused by the complexity of its first-generation predecessors and enabled multiple intervertebral fusions of the lumbar spine. Of particular note was the strong connection the PathFinder^®^ system provided between the extender assembly and the screw head that prevented the extender from accidental detachment during the procedure. However, the screw head at that time was made large in order to strengthen the connection with the extender. Currently, the PathFinder^®^ design has been renewed as the NXT^®^ system (Zimmer Biomet, Warsaw, IN), which features smaller extenders and screw heads, more variations of rod inserters, and a simplified procedure.

Following the original PathFinder^®^, it can be said that the second-generation systems represented by VIPER^®^, SEXTANT Advanced^®^, MANTIS^®^, and ILLICO SE^®^ have paved the way for establishing the PPS procedure. Despite its slightly thicker extender assembly, the SEXTANT Advanced^®^ system improved many of the problems found in the original SEXTANT^®^, and the system was unique in that it enabled a powerful reduction and was able to link to a navigation system. VIPER^®^ (Figure 1c and Figure 2a–d) and ILLICO SE^®^ (Figure 1d) were highly refined designs that were rapidly adopted worldwide, and it is no exaggeration to suggest that most extender-mounted PPS systems in the United States are modeled after these systems. In fact, many of the extender-type PPS systems introduced in Europe and the United States after 2010 were similar to VIPER^®^ and ILLICO SE^®^. In other words, these were the final and definitive forms of the second-generation PSS that can also be characterized as extender-type PPS systems.

The launch of VIPER^®^ in Asia was unusual in that VIPER^®^ was introduced for implants while VIPER 2^®^ was introduced for instruments. VIPER^®^ was developed with a novel tap system (Figure 2d) that integrated a tap and tap guard together in order to simplify dilation and reduce operation time [11]. This system was also used in the subsequent PRECEPT^®^, Associa Harp^®^/Associa Zique^®^, and IBIS^®^/Pisces^®^ systems (Figure 3 and Figure 4). The most distinctive feature of VIPER^®^ was the introduction of a double lead thread pattern for its screw. As the screw is inserted, one rotation of the screwdriver causes the screw to move forward by two rotations, shortening the screw insertion time. On the other hand, the 6 mm diameter screw tended to provide slightly poorer purchase of bone in patients with osteoporosis compared to other companies due to the steeper helix angle of the thread; however, a 7 mm diameter screw could provide sufficiently strong fixation. In addition, the development of cortical fix screws (single lead thread screw tip and double lead thread screw base) in SOLERA SEXTANT^®^ and Voyager^®^ triggered a trend for similar designs that are still used today. Later, a cortical fix screw with a comparable design was introduced in VIPER^®^, which was the only PPS system with a dual lead thread tip and a quad lead thread at the base for increased pull-out strength (data from DePuy Synthes Spine) (Figure 2b). In the VIPER^®^ system, compression and distraction were applied outside the body through the extender; thus, even though the screw head could tilt slightly, sufficient compression could still be applied with very few problems, owing to its simplicity. A similar mechanism was also introduced in the subsequent Associa Harp^®^/Associa Zique^®^ and IBIS^®^ systems. In recent systems, a compressor is installed near the screw head in the body so that the screw head does not tilt; however, this poses a drawback in that it leads to a large skin incision. On the other hand, Ballista^®^ and SpiRit^®^ were equipped with a ratcheted device that could provide near-parallel compression. Although the concept was very good, there were many problems involving the ratchet mechanism getting stuck; therefore, ratcheted devices have since been rarely introduced in PPS systems.

ILLICO SE^®^ (Figure 1d) was a particularly refined system for single-level interbody fusion. Specifically, it offered a simple and strong connection between the screw head and extender, many variations for rod inserters, and a compressor that was easy to use. MANTIS^®^ was a PPS system with a plate-type extender, which had a good field of view in the extender to facilitate easy rod placement and enable long fixation in the thoracic to lumbar vertebrae (Figure 1e). Today, the third-generation ES2^®^ is evolving as a successor and next-generation system that can be used with an integrated powered screw insertion driver while retaining the good aspects of the MANTIS^®^ system (Figure 1f). The powered driver can insert a guide wire into a defined trajectory as long as the rotation speed at the time of screw insertion is controlled, and no extra force is required. The powered driver has also been introduced in Voyager ATMAS^®^ and VIPER^®^ and could potentially be an indispensable tool for PPS insertion in the future for robotic-assisted procedures.

It should be noted that there were unique medical devices and techniques that developed independently in Japan, which helped promote the use of PPS. These devices and techniques included: S-wire^®^ (Tanaka Medical Instruments Co., Ltd., Tokyo, Japan) [12], a guidewire that can prevent anterior vertebral penetration when a PPS is inserted; LICAP method [13], a technique that allows the insertion of PPS while shortening fluoroscopic irradiation time; J-probe^®^ (Tanaka Medical Instruments Co., Ltd.), a reusable device that prevents interference; and the groove-entry technique [8], a safe method for thoracic PPS insertion. In addition, an important issue to consider is the vital role that manufacturers have played in medical education between the period of introduction and establishment of PPS techniques in Asia, including Medtronic, which introduced the first-generation system, and Abbott Spine/Japan Medical Dynamic Marketing and DePuy Synthes, which introduced the second-generation systems.

## 3. Third-Generation PPS Systems

In the third-generation PPS system, a tab was introduced to the extender, which made it possible to reduce its weight and easily remove the extender by folding the table. The earliest and most widely used system on the market was the VIPER2 X-tab^®^ system (Figure 2a–d). The simplicity of the tab and the thinness of the extender were also suitable for multi-level spinal stabilization, and together with MANTIS^®^ (Figure 1e), opened new doors for long fixation procedures. Cannulated screws for S2AI were only available for the VIPER2^®^ system (standard) at the time; thus, the system was useful in cases where fixation to the sacral bone was required. PRECEPT^®^ had a lineup of both extender-type and tab-type systems, which could be used for any pathological condition (Figure 3a,b). In particular, the tab type was excellent for deformity correction because of its strong rigidity due to the connection at the tip of the tab; however, the screw was only available in double lead thread, and the rod was only available in titanium alloy with a diameter of 5.5 mm. Therefore, it was not suitable for corrective long fixation of spinal deformities. Currently, RELINE^®^ (Figure 3c,d) has been developed as a successor, which features a lower profile, rod diameter of 5.5 or 6.0 mm, choice of titanium alloy or cobalt chrome, and an ability to use the system for long scoliosis correction. In addition, because the system shares common instrumentation for both open and PPS methods, hybrid surgery is possible for revision surgery, and the strength of the tab has also been improved. In addition, the combined use of NuVasive^®^ NVM5 neuromonitoring can accurately monitor for spinal canal deviations during needle insertions, thereby increasing safety. In addition, cobalt chrome rods have been introduced in the ES2^®^, VIPER2 X-tab^®^, Voyager^®^, and RELINE^®^ systems, which are extremely useful for correcting kyphoscoliosis in patients with degenerative lumbar scoliosis. S4 FRI/Brücken (B Braun Aesculap, Melsungen, Germany) is a system that enables ligamentotaxis and is easy to use for trauma, such as burst fractures. On the other hand, the adaptation of the system to complex pathological conditions remains an issue, and further improvements are expected for the system to serve as the first-choice PPS system for trauma.

Voyager^®^ has changed its appearance from the previous SEXTANT^®^ series with a slimmer extender, and Medtronic has launched the product as a completely new system (Figure 2e,f). Various new features were expected, such as the smallest diameter extender in its class, low profile screw head, new rod inserter based on the concept of rail and pivot rotation, improved fixation with the shape of the new SOLERA screw, and integration with navigation. In fact, there is already great improvement in the accuracy of PPS insertion due to its ability to link with O-arm^®^ navigation (Medtronic). In addition, Voyager ATMAS^®^, which will be launched in 2022, is expected to provide excellent compatibility between O-arm^®^ navigation and the MAZOR X^®^ robotic system (Medtronic) with continued development in the future.

The Bendini^®^ spinal rod bending system (NuVasive) that accompanies the PRECEPT^®^ and RELINE^®^ systems takes points at the screw head of each PPS, measures the positional information of the rod connection, and uses the resulting measurement for a dedicated rod bender. The system allows predictable and reproducible rod bending to help surgeons create patient-specific rods, often requiring a single pass. Patient-specific rods minimize forces on the screw–bone interface and prevent unnecessary preloading of the construct.

On the other hand, in terms of PPS systems that are made in Japan, Associa Harp^®^ (KYOCERA) (Figure 3e) and IBIS^®^ (ORTHO Development/ Japan Medical Dynamic Marketing) (Figure 4d) were launched in 2015, followed by Saccura^®^ (Teijin Nakashima Medical) (Figure 4c), Reng^®^/PSV^®^ (HOYA), Associa Zique ^®^ (KYOCERA) (Figure 3f), and Pisces ^®^ (ORTHO Development/Japan Medical Dynamic Marketing). In these systems, implants and tools are designed according to the body size of Asian patients. The latest technology is used with a tremendous amount of attention given to detail, producing instruments with remarkable precision and ease of use.

## 4. Fourth-Generation PPS Systems

Today, the PPS technique has become a standard procedure; however, PPS systems have continued to make progress. In order to shorten the PPS insertion time and provide a safer system compared to third-generation systems, the new fourth-generation guidewire-free systems have been developed, including the VIPER PRIME^®^ (Figure 5a,b) and Voyager ATMAS^®^ (Figure 5c,d) systems. It is a novel system that eliminates the need for common steps in conventional PPS systems such as use of Jamshidi needles, bone tunnel creation, guidewire positioning, use of a dilator, and tapping. However, there are also notable problems, such as the difficulty in re-inserting the PPS, as a large PPS with a sharp tip is inserted without creating an appropriate trajectory by a needle. Therefore, beginners are encouraged to master basic PPS procedures using a third-generation system before using a wireless fourth-generation system.

The characteristic advantage of fourth-generation PPS is that it is possible to avoid radiation exposure during screw insertion by linking the system to the O-arm^®^ navigation. In addition, it is also possible to link the system to a robotic-assistance system. Power drivers are indispensable in robotic-assisted surgery and have been introduced in the Voyager^®^ and VIPER^®^ systems. It is interesting to note that the new Voyager ATMAS^®^ uses cortical threading to minimize slippage during the use of power tools and single threading (2 mm pitch) for press-fitting (Figure 5c).

## 5. Conclusions

We described the historical evolution from the first-generation SEXTANT^®^ PPS system reported in 2001 to the current PPS systems used today. PPS systems have been developed for approximately 20 years for the purpose of improving minimally invasive techniques, safety of instrumentation, and ease of use. The third-generation PPS systems established the insertion technique, and the development of the fourth-generation PPS systems have made great strides in minimizing the number of steps in the operative procedure. In the future, PPS systems are expected to continue making use of the latest technological advancements and to develop further with the aim of ensuring greater safety, reducing operator stress, and preventing complications such as insertion errors and infection.

## Figures and Tables

**Figure 1 medicina-58-01064-f001:**
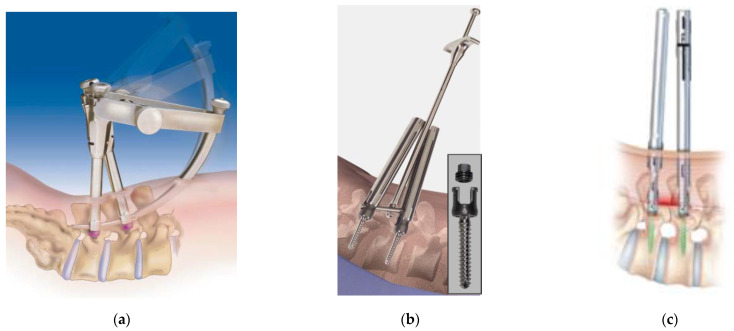
The 1st- and 2nd-generation PPS systems. SEXTANT^®^ (**a**), PathFinder^®^ (**b**), VIPER^®^ (**c**), ILLICO SE^®^ (**d**), MANTIS^®^ (**e**), and ES2^®^ (**f**).

**Figure 2 medicina-58-01064-f002:**
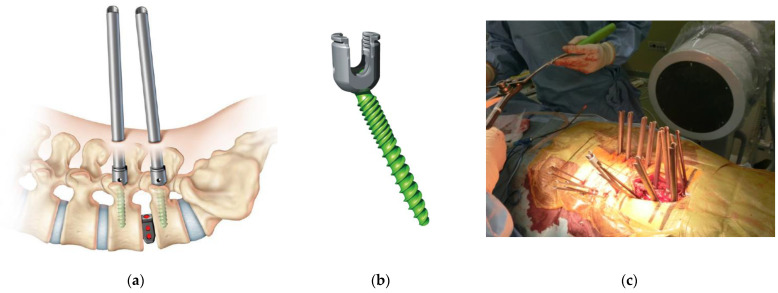
Third-generation PPS systems (VIPER2 X-tab^®^ and Voyager^®^). VIPER2 X-tab^®^ (**a**), VIPER2 cortical fix screw (tip is dual lead thread; base is quad lead thread) (**b**), VIPER2 X-tab^®^ long fixation (**c**), VIPER2^®^ tap guard (**d**), and Voyager^®^ (**e**,**f**).

**Figure 3 medicina-58-01064-f003:**
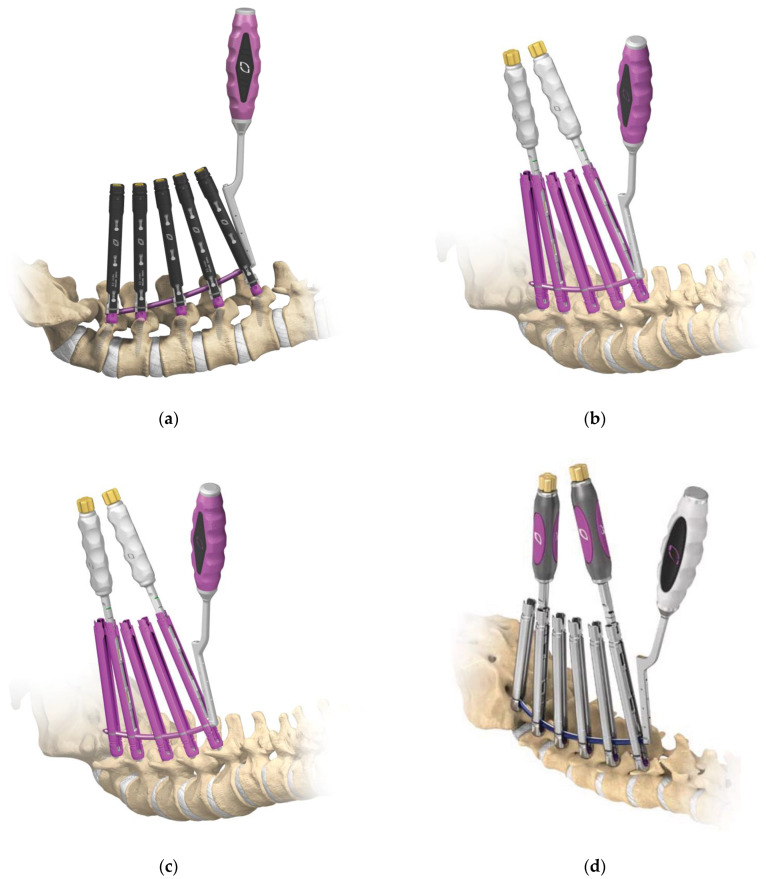
Third-generation PPS systems (PRECEPT^®^**,** RELINE^®^, Associa Harp^®^, and Associa Zique^®^). PRECEPT^®^ (extender type) (**a**), PRECEPT^®^ (tab type) (**b**), RELINE^®^ (extender type) (**c**)**,** RELINE ^®^ (tab type) (**d**), Associa Harp^®^ (**e**), and Associa Zique^®^ (**f**).

**Figure 4 medicina-58-01064-f004:**
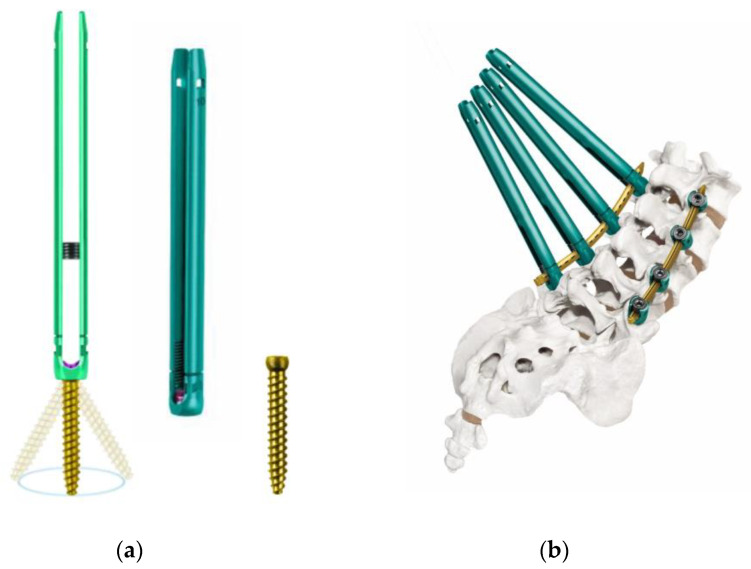
Third-generation PPS systems (CREO^®^, Saccura^®^, and IBIS^®^). CREO^®^ (**a**,**b**)**,** Saccura^®^ (**c**), and IBIS^®^ (**d**).

**Figure 5 medicina-58-01064-f005:**
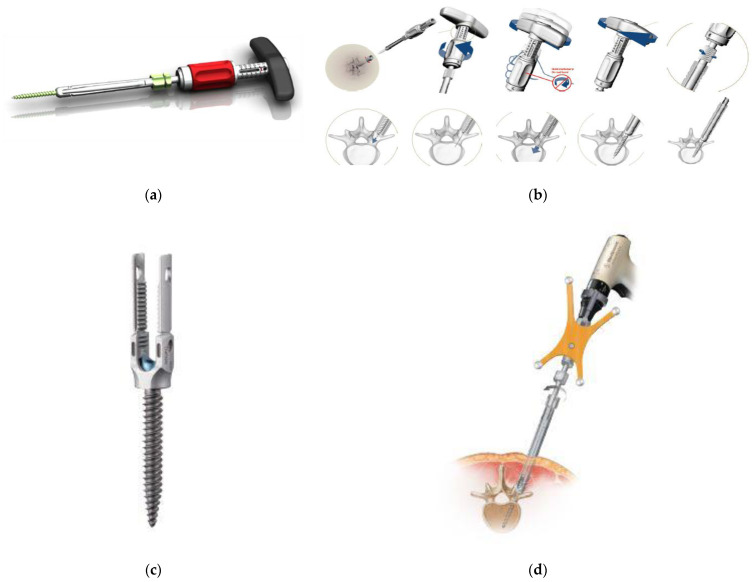
Fourth-generation PPS systems (VIPER PRIME^®^ and Voyager ATMAS^®^). VIPER PRIME^®^ (**a**,**b**) and Voyager ATMAS^®^ (**c**,**d**).

**Table 1 medicina-58-01064-t001:** Commercially Available PPS Systems.

	Year (US, JP)	System	Company
First Generation	2000,2005	SEXTANT	Medtronic
Second Generation	2007	PathFinder	Abbott Spine/Japan Medical Dynamic Marketing
	2006, 2009	VIPER	Depuy Synthes
	2009	MANTIS	Stryker
	2006, 2009	SpiRit	Synthes
	2009	Ballista	Biomet Spine
	2011	ILLICO SE	Alphatec Spine
	2012	PathFind NXT	Zimmer Spine
	2013	S4 FRI	B Braun Aesculap
	2005, 2010	SEXTANT Advanced	Medtronic
	2012, 2013	CDH SOLERA Longitude	Medtronic
	2010, 2012	CDH SOLERA SEXTANT	Medtronic
Third Generation	2008, 2012	VIPER2 system (Standard)	Depuy Synthes
	2011, 2012	VIPER2 system (X-tab)	Depuy Synthes
	2013	PRECEPT	NuVasive
	2014 (JP)	Associa Harp	KYOCERA
	2014 (JP)	IBIS	ORTHO Development/Japan Medical Dynamic Marketing
	2015	ES2	Stryker
	2015	S4 Brücken	B. Braun Aesculap
	2015, 2015	Voyager	Medtronic
	2015, 2015 2015, 2017	VIPER2 (X-tab) cortical fix screw CREO	Depuy Synthes Globus Medical
	2016	ES2	Stryker
	2016	Everest	Stryker
	2017, 2018	Voyager 5.5/6.0	Medtronic
	2019 (JP)	Saccura	Tejin Nakashima Medical
	2018 (JP)	Reng/PSV	HOYA
	2019 (JP)	Associa Zique	KYOCERA
	2019	RELINE	NuVasive
	2019	SteriSpine	KiSCO
	2021	Pisces	Japan Medical Dynamic Marketing
Fourth Generation	2017, 2018	VIPER PRIME	Depuy Synthes
	2022, 2022	Voyager ATMAS	Medtronic

## Data Availability

Not applicable.

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
