# Peer review of "The History and Development of the Percutaneous Pedicle Screw (PPS) System"

_medicina, 2022, doi:10.3390/medicina58081064_

Round 1

Reviewer 1 Report

The authors aimed to described the historical evolution of percutaneous pedicle screws. This review is well written and presentation is good

Author Response

We appreciate the reviewer’s encouraging comments. We hope that our manuscript gives the physicians significant impact.

Reviewer 2 Report

A very well written article.

The authors provided a comprehensive historical review of the use of percutaneous pedicle screws in spinal surgery.

The paper presented is well written and of considerable interest to spinal surgeons, clarifying the historical development of pedicle screw systems, the advantages and disadvantages of their various generations. 

Undoubtedly, the information presented in the article will be useful to the practitioner.

The article may be published without further correction.

Author Response

We appreciate the reviewer’s comments. We are deeply happy if the physicians could learn a lot of things from this historical review.

This manuscript is a resubmission of an earlier submission. The following is a list of the peer review reports and author responses from that submission.

Round 1

Reviewer 1 Report

The authors aimed to described the historical evolution of percutaneous pedicle screws. This review is well written and presentation is good

1. Corrections:References should be corrected because them are 13 and not 15

2. Informed Consent Statement: should be N/A

3. 

Reviewer 2 Report

Authors presented in their paper the historical development of the Percutaneous Pedicle Screw (PPS) System for treating different pathological conditions of the lumbar spine. 

They wrote about four generations of PPS systems and analyzed some improvements during the last 20 years. 

The authors considered just the technical characteristics of the PPS systems and didn't comment on treatment outcomes and other factors regarding medical benefits or complications. 

Anyway, they gave to readers a very good systematic overview of the PPS systems development.

If the paper subject (mainly the technical aspect of PPS) is in the focus of your journal it could be published. 

Best regards!